# Prediction of Nitrogen Dosage in ‘Alicante Bouschet’ Vineyards with Machine Learning Models

**DOI:** 10.3390/plants11182419

**Published:** 2022-09-16

**Authors:** Gustavo Brunetto, Lincon Oliveira Stefanello, Matheus Severo de Souza Kulmann, Adriele Tassinari, Rodrigo Otavio Schneider de Souza, Danilo Eduardo Rozane, Tadeu Luis Tiecher, Carlos Alberto Ceretta, Paulo Ademar Avelar Ferreira, Gustavo Nogara de Siqueira, Léon Étienne Parent

**Affiliations:** 1Soil Science Department, Federal University of Santa Maria, Santa Maria 97105-900, Brazil; 2Forest Science Department, Federal University of Santa Maria, Santa Maria 97105-900, Brazil; 3Fruticulture Department, State University of Paulista “Julio Mesquita Filho”, Registro 11900-000, Brazil; 4Rio Grande do Sul Federal Institute, Campus Restinga, Porto Alegre 91791-508, Brazil; 5Department of Soil and Agri-Food Engineering, Laval University, Québec City, QC G1V 0A6, Canada

**Keywords:** N fertilization, model-building, anthocyanin, total titratable acidity, vineyard management

## Abstract

Vineyard soils normally do not provide the amount of nitrogen (N) necessary for red wine production. Traditionally, the N concentration in leaves guides the N fertilization of vineyards to reach high grape yields and chemical composition under the *ceteris paribus* assumption. Moreover, the carryover effects of nutrients and carbohydrates stored by perennials such as grapevines are neglected. Where a well-documented database is assembled, machine learning (ML) methods can account for key site-specific features and carryover effects, impacting the performance of grapevines. The aim of this study was to predict, using ML tools, N management from local features to reach high berry yield and quality in ‘Alicante Bouschet’ vineyards. The 5-year (2015–2019) fertilizer trial comprised six N doses (0–20–40–60–80–100 kg N ha^−1^) and three regimes of irrigation. Model features included N dosage, climatic indices, foliar N application, and stem diameter of the preceding season, all of which were indices of the carryover effects. Accuracy of ML models was the highest with a yield cutoff of 14 t ha^−1^ and a total anthocyanin content (TAC) of 3900 mg L^−1^. Regression models were more accurate for total soluble solids (TSS), total titratable acidity (TTA), pH, TAC, and total phenolic content (TPC) in the marketable grape yield. The tissue N ranges differed between high marketable yield and TAC, indicating a trade-off about 24 g N kg^−1^ in the diagnostic leaf. The N dosage predicted varied from 0 to 40 kg N ha^−1^ depending on target variable, this was calculated from local features and carryover effects but excluded climatic indices. The dataset can increase in size and diversity with the collaboration of growers, which can help to cross over the numerous combinations of features found in vineyards. This research contributes to the rational use of N fertilizers, but with the guarantee that obtaining high productivity must be with adequate composition.

## 1. Introduction

In wine production, the grape composition is of utmost importance [1]. ‘Alicante Bouschet’ is being used as a deep purple wine to embolden red wines [2]. The nitrogen (N) fertilization can impact the biochemical composition of ‘Alicante Bouschet’ [3,4]. Excess N also increases plant vigor, the length of the growing period, the canopy density, and the grape sensitivity to fungal diseases as well as several plant traits [1,5]. Excess N reduces light penetration throughout the canopy, alters the grape biochemical composition, and contributes to N loss by leaching or runoff [3,6,7,8,9,10]. Excess N has been shown to decrease the total soluble solids (TSS) and anthocyanin content (TAC) and increase the total titratable acidity (TTA) [11,12]. The N deficiency reduces vine growth, berry set, berry yield, grape N content, and bud fruitfulness (number of cluster primordia per bud or number of clusters per shoot).

The internal N status of grapevine is traditionally diagnosed by leaf or petiole analysis at flowering or veraison [13,14]. However, the interpretation of the foliar N status is based on the calibration of tissue N concentration against the grape yield or composition in the current year. The vineyard manager must contemplate the results of tissue analysis at full bloom or veraison, which occurs after the annual fertilization is done at bud break. This happens more than one month after full bloom and more than three months post veraison, to predict final the yield. Moreover, the yield to reach high-quality grape differs from the yield potential of the vineyard [1]. Woody species such as grapevine may also show little response to nutrients added during the current production season [15,16,17,18] due to the carryover effects of nutrients and carbohydrates stored during the preceding seasons [1,7,15,19,20].

A long-term experiment is required to associate the nutrient status of perennials measured at time t to predict the stand performance at time t + 1 and adjust the fertilization in time [21,22,23]. There is a great challenge to decipher the complexity of site-specific feature combinations between geology, geomorphology, soil, climate, micro-biology, vine biology, and human interventions to make accurate predictions of the grape yield and quality and meet the production targets [7,24,25].

Machine learning (ML) models can integrate the carryover effects and other features to make predictions [26]. Predictive machine learning (ML) models have been applied with success in several domains such as marketing, banking, customer relationship management, engineering, clinical medicine, and various other areas of science to classify data, select key variables, and support decisions [27]. Compared to classical statistical models, data mining can address much more key problem-relevant patterns in the analysis [27]. The knowledge reported in sets of large and diversified data can be processed by ML tools and communicated to domain experts. In comparison, a classical approach of agricultural experimentation and data integration assumes, in a much easier fashion, that all factors but the ones being varied are equal or at optimum levels under the *ceteris paribus* assumption [28,29].

The application of ML to the analysis of agricultural data is still in its infancy. Machine learning has been used in different domains of plant science such as plant breeding [30], in vitro culture [31], stress phenotyping [32], stress physiology [33], plant system biology [34], plant identification [35], and pathogen identification [36]. Machine learning has also been used to build models of crop response to fertilization [26,37,38] and to derive nutrient standards for several fruit crops accounting or not for the carryover effects [39]. The ML models in plant nutrition often exceeded 0.8. To our knowledge, neither nutrient standards nor predictive models have been developed to account for the carryover effects of nutrients or carbohydrates in grapevine. We hypothesized that (1) nitrogen fertilization impacts foliar composition in grapevines as well as berry yield and quality, and (2) nutrients and carbohydrates stored in grapevine impacts the grape yield and quality in the following season. The aim of this study was to predict N management from local features to reach a high berry yield and quality in ‘Alicante Bouschet’ vineyards.

## 2. Results

### 2.1. Climatic Conditions and Grape Quality Indices

The season 2018/2019 was the rainiest, 2016/2017 the coldest, and 2015/2016 the warmest (Table 1). The number of chilling hours (7 °C) was the lowest in 2015/2016 (290 h). As shown by the Shannon distribution index (SDI), rainfall was more evenly distributed in 2015/2016, 2017/2018, and 2018/2019 than in other years, but was adjusted to crop need by irrigation. The berry yield varied between 3 and 34 t ha^−1^ during the experimental period. Foliar N concentrations and stem diameter also varied widely at full bloom and veraison (Table 2).

### 2.2. Machine Learning Model-Building and Foliar N

Features showing the highest RRelief scores on target climate, N doses, foliar N, stem diameter, and application mode after accounting for the carryover effects. Other features showed a negligible impact. The R^2^ value of the regression ML models depends on the target variable (Table 3). All of the quality attributes showed high R^2^ values. We selected TAC as the quality attribute because ‘Alicante Bouschet’ was used to embolden red wines. The classification accuracy (CA) was highest using 14 t ha^−1^ as the cutoff yield (CA = 0.860) and 3900 mg L^−1^ as the TAC cutoff content (CA = 0.936).

Nitrogen concentration quartiles at full bloom and veraison were compared to the literature data (Table 4). The N concentration quartiles accounting or not for the carryover effects differed slightly from the N concentration ranges reported in the literature. The N concentration quartile ranges were higher for berry yield than TAC. The tradeoff at a centroid N concentration of 24 g N kg^−^^1^ between high berry productivity and high TAC depended slightly on the carryover effects (current-year vs. preceding-year) The N concentration centroids for TAC were 21–23 g N kg^−^^1^ at full bloom and 22–24 g N kg^−^^1^ at veraison. The N concentration centroids at high berry yield were 27 g N kg^−^^1^ at full bloom and 23–24 g N kg^−^^1^ at veraison. While the centroid foliar N differed to reach a high berry yield and quality at full bloom, it was similar at veraison, typically occurring 3–4 weeks before harvest. The N concentration ranges reported in the literature also varied more widely at full bloom that at veraison.

### 2.3. Predictions

The goal of the predictive model was to relate the grape yield and quality to the N dosage in order to assist the growers’ decision on the most appropriate N application rate in the following year. The most accurate ML model depended on the target variable (Table 4). Except for the must pH, the predictions generally paralleled the actual target variables despite occasional differences in the data distribution (Figure 1), likely attributable to the ‘year’ effect. Must pH was poorly predicted. Indeed, 2019 was the rainiest. The N dosage to reach high berry yield and quality varied between 0 and 40 kg N ha^−1^.

## 3. Discussion

### 3.1. Machine Learning Model-Building

The present plant nutrition diagnostic methods based on statistical relationships between the nutrient status and crop performance are contemplative. Indeed, the results of the current-year tissue analysis arrived after the application of N fertilization and without any hint on the most probable final grape yield or must composition. In this paper, ML models were elaborated to predict the grape yield and composition. The selected learners processed combinations of key features. This differed from conventional research that addressed every feature separately under the *ceteris paribus* assumption across other features [29].

To produce high-quality wine and meet the market demand, managers of vineyard agroecosystems should be able to test and rank key features impacting the grapevine yield and quality to predict the nutrient requirements and stand performance in advance to adjust the fertilization programs. The prediction should integrate the carryover effects from nutrient and carbohydrate storage in perennials. Tissue analysis, stem diameter, and berry yield measured during the preceding season(s) were used as indicators for stored nutrients and carbohydrates to adjust the fertilization program well in advance of fertilizer applications.

Easily documented features were ranked for their impacts on berry yield and composition, then combined to predict the N dosage and we set apart the best performing specimens. The key features of high-performing specimens form the basis to assist decisions based on the documented facts. The manager can compare the features of a farm specimen to those of high-performing specimens to focus on the ones that appear to be limiting. Hence, ML models allow for the testing of new hypotheses on the way to improve stand performance.

Machine learning models are well-suited to unravel the complexity of agroecosystems [44]. Machine learning models can integrate the scientific knowledge from the literature as well as observable data to make predictions [45]. Cross-validation, as used in this paper, is a means to initiate model-building. Because the model must be further validated on new data, a larger and more diversified dataset could be acquired across several grapevine agroecosystems to avoid model overfitting and enhance model reliability.

### 3.2. Tissue Test at High Grape Yield and Quality Levels

Sandy soils showing low organic matter content usually do not provide enough mineral N to meet the demand of grapevines, especially for high-yield cultivars such as ‘Alicante Bouschet’ [3,4]. Tissue N concentration of ‘Alicante Bouschet’ was thus impacted by N fertilization. As anticipated, there was a tradeoff of foliar N concentrations to achieve high productivity or color intensity experimental results in the ‘Alicante Bouschet’ vineyard (Table 3). The foliar N range at full bloom resembled that of [42] for TAC, and those of [41] as well as [40] for grape yield. The N concentration quartiles at veraison for grape yield and TAC were comparable and resembled the N ranges presented by [1] for Europe.

Only foliar N was analyzed during this experimentation. Tissue N content measured at full bloom or veraison informs on N metabolism until veraison [1]. It is currently diagnosed against critical nutrient concentration levels or ranges [46,47]. However, nutrient ranges may lead to the wrong diagnosis as it is biased by pairwise and high-level nutrient interactions [37,48]. The interactive interpretation of the foliar N status of grapevine was first illustrated by the NPK ternary diagram or using pairwise N/P and N/K ratios [1]. While nutrient interactions are generally expressed as pairwise ratios [49], high-order interactions can also be reported as isometric log ratios or balances [50]. Centered log ratios can integrate pairwise ratios [48,50] and provide nutrient standards for grapevine [40]. Such an approach should also be addressed in future research for the carryover effects.

Similar to grapevine, cranberry has a vine growing pattern. The diagnostic tissue is composed of vegetative and reproductive runners, hence leaves and woody stems [51] that can store carbohydrates and nutrients. Such a sampling procedure can allow for integrating the yearly and carryover effects to make predictions on the N requirements [26]. Tissue nutrient diagnosis of grapevine can thus be made more accurate by including the carryover effects as well as visual indicators such as cultivar, production objectives, plant morphology, vigor, canopy density, floral induction, bud fruitfulness, and yield potential [1]. Although not analyzed in the present study, grape N content at harvest could also provide an integrative view of plant N status across the entire season including ripening.

### 3.3. Grape Yield and Quality

Nitrogen applications can increase TTA in the must because of an increase in the vigor of organs such as the leaves and branches of the year, reducing the incidence of sunlight on clusters, which delays the degradation of acids in berries [13]. The increase in must pH can also occur because larger leaf areas increase the transpiration and absorption of water and nutrients such as K, which accumulates in berries [52,53]. The K accumulation takes place because of the intense cell division and elongation in the tissues of the berries. The K in the berry can promote stoichiometric exchange with protons (H^+^), forming K bitartrate, which increases the must pH. The malic:tartaric acid ratio may also decrease, further contributing to the pH increase [52].

The maintenance or increase of must TSS, even at high production levels, may occur in grapevines because of a larger number of clusters with either fewer berries or berries of smaller diameter that contain higher concentrations of sugars [3,54]. However, some compounds in the must such as TPC can be diluted, where berries show a larger diameter [55,56]. This is not always the case for TAC, especially for cultivars such as ‘Alicante Bouschet’, which shows a high TAC in berry, must, and wine [57]. Normally, grapevines grown in sandy soils and fertilized with N absorb a proportion of the added N that promotes plant vigor and decreases the activity of enzymes regulating the synthesis of compounds such as TAC due to reduced sunlight [58,59].

### 3.4. Carryover Effects of Fertilization

Roots can store ≈75% of the N reserves of grapevines that has been acquired from the leaves before leaf fall [1] as a small percentage of added N is recovered by the grapevine within the season of fertilizer application [42,60]. Indeed, urea–N transferred preferentially to the leaves contributed 3–8% to the nutrition of plant organs [42], indicating low N-use efficiency in the year of fertilizer application and high potential for carryover effects. The rye green manure left on the soil surface (2 t dry matter ha^−^^1^ providing 82 kg N ha^−^^1^) was also shown to contribute less than 2% to biomass N in the annual and perennial grape organs 28 weeks following fertilizer application [60]. Previous N fertilization can contribute later to N supply through microbial N transformation into organic N, followed by the mineralization of organic N [8,61].

Considering the carryover effects, the predictive model indicated that a grape yield of 15 t ha^−^^1^ could be obtained in 2019 across N doses. The state guidelines for testing soils low in organic matter content (<2.5%) would be 50 kg N ha^−^^1^ for this 8-year-old vineyard [14]. Considering foliar N at full bloom (1.7–2.0%), the state recommendation would be more restrictive at 20 kg N ha^−^^1^ for the yield level of 15 t ha^−^^1^. Despite similarities with the state guidelines, the suggested N dosage in the range from 0 to 40 kg N ha^−^^1^ differentially impacted the grape quality indices, providing more information on the potential effect of various N fertilization scenarios on the grape quality.

The decision to skip or adjust N fertilization in the following year must consider not only the responses to yearly growth-impacting factors, but also the carryover effects. However, 2019 was much rainier compared to other years, and the effect of high rainfall on the grape yield and quality was not within the range of the 2015–2018 calibrated model. Additional experimental and observational data are thus necessary to cross over more climatic indices and compare the predictions of grape yield and quality under various climatic conditions. This is even more necessary in vineyards located in countries where climate change is more intense.

### 3.5. Building a Dataset for Nutrient Management of Vineyards

Fertilization is needed to sustain grapevine internal nutrient reserves and yields [62,63]. This requires conducting fertilizer experiments under various sets of uncontrollable and controllable factors. In Brazil, grapevine productivity has been tested in response to N fertilization and irrigation [3,4,13,42,60], P fertilization [64], and K fertilization [65,66,67] to develop state fertilizer recommendations. Nevertheless, growers must also rely on their own experience and expertise to manage the fertilization of vineyards.

The different pathways of the carryover effects detected by the soil and tissue tests show the importance of selecting successful combinations of uncontrollable and controllable factors to achieve high grape yield and quality. This requires documenting numerous combinations of large and diversified experimental and observational (growers) datasets. In addition to experimental data such as those collected in this research and in other experiments [42,60,62], the observational data of the growers must be collected reliably and ethically [40] to support information-based models of artificial intelligence. The datasets should be built uniformly to facilitate knowledge exchange among stakeholders with the objective of improving local fertilizer recommendations from well-documented successful combinations of local features.

## 4. Materials and Methods

### 4.1. Experimental Setup

A 5-year study was conducted during the 2015–2019 period in an ‘Alicante Bouschet’ vineyard established in 2011 in Santana do Livramento (30°48′31″ S; 55°22′33″ W), Rio Grande do Sul State, Brazil. The soil was classified as Typic Hapludalf [68]. The climate of the region is humid subtropical (Cfa according to Köppen climate classification) [69], characterized by mild temperatures and regular precipitations throughout the year. Meteorological data were obtained from the National Institute of Meteorology [70]. Land topography was slightly undulating.

At the beginning of the experiment, the 0–20 cm soil layer showed the following characteristics [71]: pH of 5.5 in water (1:1 ratio), 25 mg P kg^−1^, and 72 mg K kg^−1^ extracted using the Mehlich-1 method, and 0.0 cmol_c_ Al kg^−1^, 1.99 cmol_c_ Ca kg^−1^, and 0.92 cmol_c_ Mg kg^−1^ extracted using the KCl 1 N method. The soil contained 822 g sand kg^−1^, 115 g silt kg^−1^, 63 g clay kg^−1^, and 11 g Walkley–Black organic matter kg^−1^. Prior to the experiment, the vineyard received 45 kg P_2_O_5_ ha^−1^ as triple superphosphate (41% P_2_O_5_) and 37.5 kg K_2_O ha^−1^ as KCl (60% K_2_O) [71].

Cultivar ‘Alicante Bouschet’ was grafted on the ‘Paulsen 1103′ rootstock. The training system was ‘espalier’ with double string on the first thread. Winter pruning retained a comparable number of buds across treatments. The prunings were exported out of the field as a preventive measure against plant diseases. The spacing was 2.8 m between rows and 1.2 m between plants, for a plant density of 2976 plants ha^−1^. The vegetation between rows was composed of *Paspalum notatum*, *Paspalum plicatulum*, and *Lolium multiflorum* mowed mechanically at a 10 cm height up to five times during the annual cycle of the grapevine. The vegetation on the planting rows was desiccated annually using herbicides. The dates of events in the experimental vineyard are reported in Table 5.

### 4.2. Treatments

The N treatments were 0, 20, 40, 60, 80, and 100 kg N ha^−1^, surface-applied as urea 14 d after budbreak. There were three methods of N application: (1) conventional—granular form without mechanical incorporation and without irrigation within the crown projection; (2) conventional-irrigated—granular form without mechanical incorporation within the crown projection and drip irrigation for 30 min d^−1^ after fertilizer application; and (3) fertigation for 30 min d^−1^—liquid form without mechanical incorporation within the crown projection. More details on the treatments are provided in [4]. There were five replications per treatment. With six N treatments, three N application methods, five replications, and five years of experimentation, there were 450 observations in total.

### 4.3. Foliar and Fruit Analysis

Six fully expanded leaves opposite to the cluster in the middle third of the annual growth [14] were collected at full bloom from October to November and at veraison (fruit ripening), where berries changed color from December to January [1,72,73]. The stem diameter was also measured at full bloom and veraison. Leaves were washed gently with distilled water, oven-dried at 65 °C, and ground to less than 2 mm. Leaves were acid-digested [71]. Total N was quantified by semi-micro Kjeldahl. Berries were harvested in February and the yield was reported as kg ha^−1^. Total soluble solids (TSS), total titratable acidity (TTA), pH, total anthocyanin content (TAC), and total phenolic content (TPC) were analyzed as described in [4].

### 4.4. Meteorological Data

Meteorological data were obtained from the closest weather station [70]. Climatic indices were synthesized as growing degree-days [19], number of chilling hours, rainfall indices, days of clear weather, and the occurrence of catastrophic events (hail, frost, …). While mild temperatures and moderate rainfall promote N uptake by the grapevine, cool, cloudy, and rainy seasons may reduce N uptake and photosynthesis and lead to N leaching [1].

The optimum temperature for grapevine ranges between 10 and 35 °C [1]. The number of cumulated growing degree days (GDD) was computed using Model 1 in [74], as shown in Equation (1):(1)$$GDD=\sum_{i=1}^{t}\left(T_{min}+T_{max}\right)/2-T_{base},$$
where *i → t* represents the length of the period; $\left(T_{min}+T_{max}\right)/2$ is the daily mean temperature averaged between minimum $T_{min}$ and maximum ($T_{max}$) temperatures; and $T_{base}$ is the base temperature set at 10 °C [75]. Cumulated days and degree-days were computed from bud break to harvest. The GDD indicated that the region was classified as Region III according to the Winkler index. The Shannon diversity index (SDI) is a measure of rainfall distribution [76]. The SDI was computed between bud break and harvest as shown in Equation (2):(2)$$\;SDI=-\left[\sum_{i=1}^{t}ln\left(p_{i}\right)\right]/ln\left(t\right)$$
where $p_{i}$ is the fraction of daily rainfall relative to total rainfall between bud break and harvest and *t* is the number of days between bud break and harvest; *SDI* → 1 implies a trend toward rainfall evenness (i.e., equal amounts of rainfall in each day); *SDI* → 0 implies a trend toward complete unevenness where all rain would fall in a single day.

### 4.5. Statistical Analysis

Machine learning (ML) models were those available in the Orange data mining freeware vs. 3.29.3. Models were tested in regression (continuous variable) or classification (categorial variable) modes. Categories were set about a specified target for yield or must composition. Models were trained using stratified cross-validation (k = 10). The features and target variables are presented in Table 6.

Relief results are commonly viewed as selection methods for feature subsets to guide the machine learning models [77]. Relief family algorithms are robust and noise tolerant ranking estimators to detect conditional dependencies between attributes. We used gain ratio (classification mode and RRelief (regression mode) as Relief estimators.

The confusion matrix sets apart true negative (quadrant for high predicted and high actual targets), true positive (low predicted and low actual targets), false negative (high predicted and low actual targets), and false positive specimens (low predicted and high actual targets). The nutrient standards for the target variable were computed as quartiles among the true negative specimens.

The current-year ML model was elaborated in classification mode to compute the statistics for the current-year relationships between target variable and features as follows in Equation (3) [40]:(3)$$Target_{t}=f\left(Features_{t}\right),$$

The carryover ML regression model included previous foliar N at full bloom or veraison, berry yield, biochemical components, and stem diameter at veraison. The carryover model was elaborated in Equation (4):(4)$$Target_{t+1}=f\left(Features_{t}\right),$$

The carryover dataset comprised the first four years of observations (2015–2018). The predicted response models used the last year of observations (2019). As a result, there were three datasets to run ML models: current-year (450 observations in total), carryover (360 observations in 2015–2018), and prediction (180 observations in 2019).

## 5. Conclusions

This study addressed the N management of ‘Alicante Boschet’ as a blending ingredient for red wines. Predictive ML models estimated the variables from key features. The model could adjust the N dosage to the internal plant reserves of nutrients and carbohydrates. Using foliar N concentration required to reach a high grape yield and quality of ‘Alicante Bouschet’, local (0–40 kg N ha^−^^1^) and state recommendations (20 or 50 kg N ha^−^^1^) may differ depending on the selected target variable. The dataset must be enhanced to include many more and diversified experimental and observational data to cross over the numerous combinations of features impacting the target variables and to make accurate predictions across various climatic and soil conditions. This research contributes to the rational use of N fertilizers to reach high productivity and adequate must composition under site-specific conditions.

## Figures and Tables

**Figure 1 plants-11-02419-f001:**
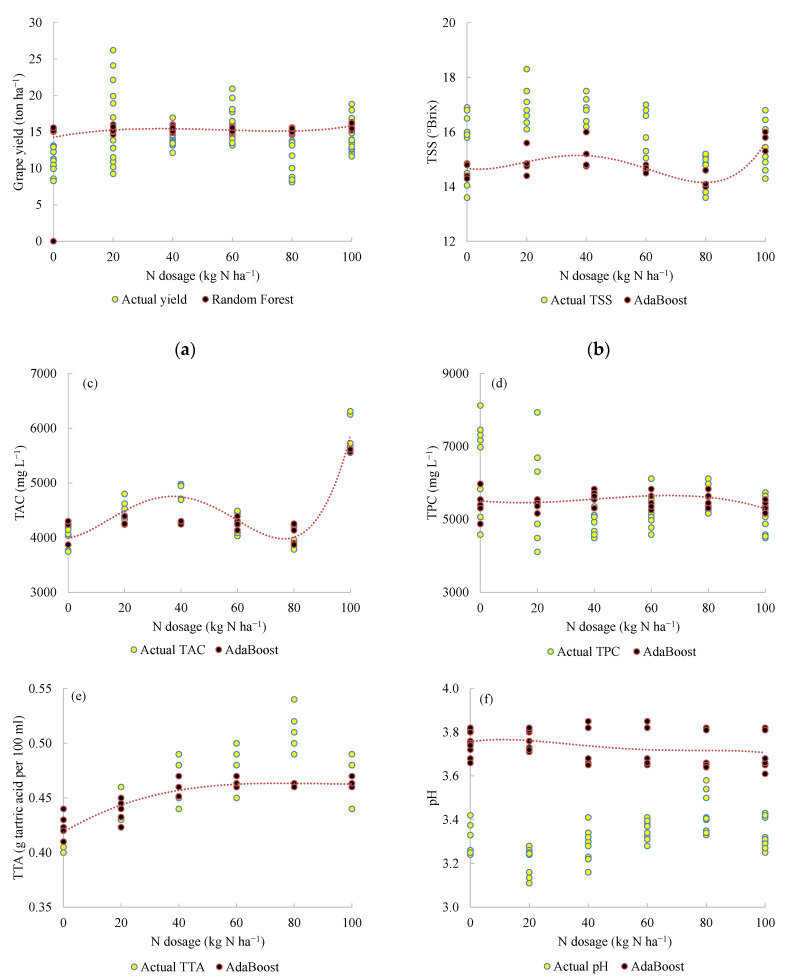
Prediction of the yield (**a**), total soluble solids (TSS) (**b**), total anthocyanin concentration (TAC) (**c**), total polyphenol concentration (TPC) (**d**), total acidity concentration (TTA) (**e**), and pH (**f**) of ‘Alicante Bouschet’ in 2019 for the model calibration for the 2015–2018 period.

**Table 1 plants-11-02419-t001:** The median values of the synthetic climatic indices and target variables during the five seasons.

Year	2014/2015	2015/2016	2016/2017	2017/2018	2018/2019
	Climatic indices				
Precipitations (mm)	722	896	592	794	1328
Shannon distribution index	0.635	0.590	0.629	0.594	0.585
Cumulated degree-days (10 °C)	1602	1927	1481	1816	1833
Number of chilling hours (7 °C)	386	290	626	342	577
Frost events (number)	27	15	37	15	22
Hail events (number)	2	3	0	1	4
Clear weather (days)	182	174	184	151	207

**Table 2 plants-11-02419-t002:** The minimum, median, and maximum values of the features and target variables during the five experimental seasons.

Variable	Unit	Minimum	Median	Maximum
Foliar N at full bloom	g N kg^−1^	15.4	26.4	33.0
Foliar N at veraison	g N kg^−1^	15.8	23.6	56.0
Stem diameter at full bloom	cm	1.9	3.9	5.5
Stem diameter at veraison	cm	2.2	4.2	25.0
Grape yield	t ha^−1^	2.74	16.39	33.75
Must total titratable acidity (TTA)	g tartric acid (100 g)^−1^	0.35	0.63	2.31
Must pH	unitless	2.80	3.61	4.30
Must total soluble solids (TSS)	°Brix	11.2	14.5	18.3
Must total phenolics content (TPC)	mg L^−1^	2470	5879	17,197
Skin total anthocyanin content (TAC)	mg L^−1^	1634	3448	6312

**Table 3 plants-11-02419-t003:** The foliar N ranges at full bloom and veraison using the yield and TAC as the target variables and comparison with the ranges reported in the literature.

Target	Method	Minimum	Centroid	Maximum	Source
		g N kg^−1^			
At full bloom					
TAC	Quartiles	19	21	24	This study—current-year TAC
TAC	Quartiles	21	23	25	This study—next-year TAC
Yield	Quartiles	24	27	29	This study—current-year yield
Yield	Quartiles	26	27	28	This study—next-year yield
Yield	Range	24	27	30	[40]
Yield	Range	24	-	30	[41]
-	Range	16	-	24	[42]
-	Range	30	-	35	[43]
At veraison					
TAC	Quartiles	22	24	26	This study—current-year TAC
TAC	Quartiles	20	22	25	This study—next-year TAC
Yield	Quartiles	22	24	25	This study—current-year yield
Yield	Quartiles	21	23	24	This study—next-year yield
-	Range	20	-	23	[1]

**Table 4 plants-11-02419-t004:** The most accurate machine learning models to predict the target variables.

Target	Adaboost		Gradient Boosting		Random Forest	
	RMSE	R^2^	RMSE	R^2^	RMSE	R^2^
Yield	3.5	0.645	3.6	0.633	3.5	0.654
TSS	0.52	0.819	0.56	0.786	0.58	0.768
TAC	260	0.943	351	0.895	398	0.866
TPC	940	0.903	1094	0.869	1142	0.857
TTA	0.062	0.864	0.062	0.867	0.061	0.869
pH	0.050	0.947	0.056	0.933	0.056	0.933

**Table 5 plants-11-02419-t005:** The dates (month/day/year) of events in the experimental vineyard.

Event	2014/2015	2015/2016	2016/2017	2017/2018	2018/2019
Bud break	09/10/2014	08/28/2015	09/10/2016	08/30/2017	08/29/2018
First N application	10/12/2014	09/20/2015	09/30/2016	10/16/2017	10/092018
Leaf sampling at full bloom	11/22/2014	11/15/2015	11/06/2016	11/28/2017	11/21/2018
Leaf sampling at veraison	01/18/2015	01/22/2016	01/09/2017	01/24/2018	n.a.
Grape harvest	02/12/2015	02/24/2016	02/16/2017	02/20/2018	02/21/2019

n.a.—not analyzed.

**Table 6 plants-11-02419-t006:** A list of the variables during the experimental years.

Variable	2015	2016	2017	2018	2019
	Features				
N dose	X	X	X	X	X
Fertilization method	X	X	X	X	X
Foliar N at flowering	X	X	X	X	X
Foliar N at veraison	X	X	X	X	X
Stem diameter at flowering	X	X	X	X	X
Stem diameter at veraison	X	X	X	X	X
Total rainfall	X	X	X	X	X
Cumulated degree-days	X	X	X	X	X
	Target variables				
	Grape yield indices				
Yield per plant	X	X	X	X	X
Yield per hectare‡	X	X	X	X	X
	Grape quality indices				
Must acidity	X	X	X	X	X
Must pH	X	X	X	X	X
Must total soluble solids	X	X	X	X	X
Must total phenolics	X	X	X	X	X
Skin total anthocyanin content	X	X	X	X	X

## Data Availability

The datasets generated during the current study are available from the corresponding author on reasonable request.

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
