# Peer review of "Prediction of Nitrogen Dosage in ‘Alicante Bouschet’ Vineyards with Machine Learning Models"

_plants, 2022, doi:10.3390/plants11182419_

Round 1

Reviewer 1 Report

In the current manuscript, Brunetto et al. tried to predict N management accurately from local features to reach high berry yield and quality in ‘Alicante Bouschet’ vineyards by using ML. Although the topic is attractive, there are some concerns that should be addressed.

-There are some typographical and grammatical errors.

- L 76: please provide some examples of machine learning in plant science. I suggest the following statement:

Machine learning has been widely used in different fields of plant science such as plant breeding (https://doi.org/10.1016/j.isci.2020.101890), in vitro culture (https://doi.org/10.1007/s00253-020-10888-2), stress phenotyping (https://doi.org/10.1016/j.tplants.2015.10.015), stress physiology (https://doi.org/10.1371/journal.pone.0240427), plant system biology (https://doi.org/10.1007/s00253-022-11963-6), plant identification (https://doi.org/10.1016/j.compag.2016.07.003), and pathogen identification (https://doi.org/10.1094/MPMI-08-18-0221-FI).

- Discussion should be improved.

- The conclusion section is very short. At least it should discuss more future work

Author Response

Response to Reviewer 1 Comments

In the current manuscript, Brunetto et al. tried to predict N management accurately from local features to reach high berry yield and quality in ‘Alicante Bouschet’ vineyards by using ML. Although the topic is attractive, there are some concerns that should be addressed.

-There are some typographical and grammatical errors.

Response: We thank the reviewer for the comments and suggestions mentioned above. The manuscript was revised.

L 76: please provide some examples of machine learning in plant science. I suggest the following statement:

Machine learning has been widely used in different fields of plant science such as plant breeding (https://doi.org/10.1016/j.isci.2020.101890), in vitro culture (https://doi.org/10.1007/s00253-020-10888-2), stress phenotyping (https://doi.org/10.1016/j.tplants.2015.10.015), stress physiology (https://doi.org/10.1371/journal.pone.0240427), plant system biology (https://doi.org/10.1007/s00253-022-11963-6), plant identification (https://doi.org/10.1016/j.compag.2016.07.003), and pathogen identi-fication (https://doi.org/10.1094/MPMI-08-18-0221-FI).

Response: We are grateful that the reviewer has verified this. The text was revised and information were added.

Discussion should be improved.

Response: The discussion was improve. Also information were added.

The conclusion section is very short. At least it should discuss more future work.

Response: The conclusion was revised.

Reviewer 2 Report

Machine  Learning (ML) is being used in agriculture and horticulture for several years. Crop yield prediction is one of the challenging problems in precision agri- and horticulture, and many models have been proposed and validated so far. This problem requires the use of several datasets since crop yield depends on many different factors such as climate, weather, soil, use of fertilizer. This indicates that crop yield prediction is not a trivial task; instead, it consists of several complicated steps.
Prescribing optimal nutrient doses is also challenging because of the involvement of many variables including weather, soils, land management, genotypes, and severity of pests and diseases. Where sufficient data are available, machine learning algorithms can be used to predict crop performance. The objective of this study was to determine an optimal  ML model predicting nitrogen requirements for berry yield and quality in ‘Alicante Bouschet’ vineyards. The Authors exploited a data set of 450 observations conducted for five years (2015-2019) in Santana do Livramento (Brazil). All parts of the manuscript is interesting and clearly summarize data valuable for the research community.

GENERAL COMMENTS:
TITLE
The paper title is well stated, it is informative and concise.

ABSTRACT, INTRODUCTION
Abstract needs improvement (results are described too detailed).
The introduction was not well written, and it is too briefly presenting the subject and research problem. There is a lot of data in the literature- Please describe the application of ML.

MATERIAL AND METHODS
Material and research methods are presented appropriately. Experimental setup and the description in the methods section are well structured, and the statistical analysis is done alright. I have one corrective remark:
> 4.2. Treatments: This section lacks detail and is confusing. Please describe the soil profile more informatively.

RESULTS
The results obtained in this study are interesting. Results presented correctly.

DISCUSSION
In general, the discussion of the results and conclusions are correct, but not sufficient. The topic was not well discussed. The authors do not make fully use of the literature resources.

CONCLUSIONS
Repeated abstract of the work. This part needs to be improved.

The text of the manusctipt is not formatted correctly yet. Please verify the correctness of the literature and make a linguistic correction of the text by native speaker.

Author Response

Response to Reviewer 2 Comments

Machine  Learning (ML) is being used in agriculture and horticulture for several years. Crop yield predic-tion is one of the challenging problems in precision agri- and horticulture, and many models have been proposed and validated so far. This problem requires the use of several datasets since crop yield depends on many different factors such as climate, weather, soil, use of fertilizer. This indicates that crop yield prediction is not a trivial task; instead, it consists of several complicated steps.

Prescribing optimal nutrient doses is also challenging because of the involvement of many variables in-cluding weather, soils, land management, genotypes, and severity of pests and diseases. Where sufficient data are available, machine learning algorithms can be used to predict crop performance. The objective of this study was to determine an optimal  ML model predicting nitrogen requirements for berry yield and quality in ‘Alicante Bouschet’ vineyards. The Authors exploited a data set of 450 observations conduct-ed for five years (2015-2019) in Santana do Livramento (Brazil). All parts of the manuscript is interesting and clearly summarize data valuable for the research community.

GENERAL COMMENTS:

TITLE

The paper title is well stated, it is informative and concise.

Response: the title stayed in the manuscript.

ABSTRACT, INTRODUCTION

Abstract needs improvement (results are described too detailed).

Response: The abstract was revised and information were added.

The introduction was not well written, and it is too briefly presenting the subject and research problem. There is a lot of data in the literature- Please describe the application of ML.

Response: The text was revised and information were added.

MATERIAL AND METHODS

Material and research methods are presented appropriately. Experimental setup and the description in the methods section are well structured, and the statistical analysis is done alright. I have one corrective remark:

> 4.2. Treatments: This section lacks detail and is confusing. Please describe the soil profile more in-formatively.

Response: Information about the soil profile were added.

RESULTS

The results obtained in this study are interesting. Results presented correctly.

DISCUSSION

In general, the discussion of the results and conclusions are correct, but not sufficient. The topic was not well discussed. The authors do not make fully use of the literature resources.

Response: The topic discussion was revised and information were added.

CONCLUSIONS

Repeated abstract of the work. This part needs to be improved.

Response: The topic was revised.

The text of the manuscript is not formatted correctly yet. Please verify the correctness of the literature and make a linguistic correction of the text by native speaker.

Response: The manuscript was formatted and the text was revised.

Round 2

Reviewer 1 Report

Unfortunately, my comments have not been addressed.

The revised manuscript is not at the level that it draws the interest of others in the field. I see as a strong weakness that the paper actually contains two messages which are not well separated: One about the application of machine learning and a second one about the results of an experiment in which this method was applied. The paper does not seem to separate these messages very well, actually skips over the method in a much too easy way and mostly discusses some biological results rather than the pro's and cons and alternatives of the method. The manuscript contains serious flaws.

The introduction was not written in an informative manner and was so weak.

The objectives considered in this study were not clearly addressed.

The paper is about the application of machine learning in precision nitrogen management. However, there is no information about machine learning in the introduction part. As previously mentioned, the authors should discuss the advantages and challenges of machine learning in this field and provide some examples of the robustness of machine learning in other fields.

Machine learning has been widely used in different fields of plant science such as plant breeding (https://doi.org/10.1016/j.isci.2020.101890), in vitro culture (https://doi.org/10.1007/s00253-020-10888-2), stress phenotyping (https://doi.org/10.1016/j.tplants.2015.10.015), stress physiology (https://doi.org/10.1371/journal.pone.0240427), plant system biology (https://doi.org/10.1007/s00253-022-11963-6), plant identification (https://doi.org/10.1016/j.compag.2016.07.003), and pathogen identification (https://doi.org/10.1094/MPMI-08-18-0221-FI).

The discussion should be completely rewritten to answer the following question:

Why do we need to use machine learning models? Which parts are required to be defined by an automated approach (evolutionary algorithm) instead of using trial and error?  The answer to this question is an overlooked part that needs to be discussed there.

It is not interesting for readers to know somebody conducted research on some parameter somewhere.  When they want to refer to a study, it is important to know how it could be related to the objectives of the current study. For example, the authors were talking about the advantages of data-driven models in precision nutrient management. So, we need some citations supporting this statement. For example, why authors utilized machine learning, what the size and the scale of the problem, what was the results, did they compare ML approach with other approaches in terms of complexity, speed, accuracy, what they found, etc. It should be short but also it should be informative as well.

Author Response

Response to Reviewer 1 Comments

Unfortunately, my comments have not been addressed.

The revised manuscript is not at the level that it draws the interest of others in the field. I see as a strong weakness that the paper actually contains two messages which are not well separated: One about the application of machine learning and a second one about the results of an experiment in which this method was applied. The paper does not seem to separate these messages very well, actually skips over the method in a much too easy way and mostly discusses some biological results rather than the pro's and cons and alternatives of the method. The manuscript contains serious flaws.

The introduction was not written in an informative manner and was so weak.

Response: The introduction was rewritten to focus on the importance of considering several features and carryover effects on target variables. Predictive Machine Learning methods are alternatives to traditional contemplative methods.

The objectives considered in this study were not clearly addressed.

Hypothesis and objective were rewritten.

Response: We hypothesized that berry yield and composition can be predicted from N fertilization, foliar N composition, carryover effects and other key features. The aim of this study was to assess the foliar N concentration and N dosage most appropriate to reach high berry yield and quality of ‘Alicante Bouschet’ using ML predictive models.

The paper is about the application of machine learning in precision nitrogen management. However, there is no information about machine learning in the introduction part. As previously mentioned, the authors should discuss the advantages and challenges of machine learning in this field and provide some examples of the robustness of machine learning in other fields.

Response: See l. 67-104 in the actual paper form.

The discussion should be completely rewritten to answer the following question:

Why do we need to use machine learning models? Which parts are required to be defined by an automated approach (evolutionary algorithm) instead of using trial and error?  The answer to this question is an overlooked part that needs to be discussed there.

Response: We added a section l. 154-184 to support ML as alternative method in grapevine N management.

It is not interesting for readers to know somebody conducted research on some parameter somewhere.  When they want to refer to a study, it is important to know how it could be related to the objectives of the current study. For example, the authors were talking about the advantages of data-driven models in precision nutrient management. So, we need some citations supporting this statement. For example, why authors utilized machine learning, what the size and the scale of the problem, what was the results, did they compare ML approach with other approaches in terms of complexity, speed, accuracy, what they found, etc. It should be short but also it should be informative as well.

Response: The comment was important. See l. 67-104 and 154-184. Also, features can be processed by ML models to avoid discarding key features under the ceteris paribus assumption. Carryover effects as well as other key features are based on research and feature ranking.

Reviewer 2 Report

This paper by  Brunetto et al. has clearly benefited from the revision, as advised by reviewers.

Author Response

We appreciate your comments

Round 3

Reviewer 1 Report

All my comments have been addressed. I think that the current version of the MS can be published in Plants.

Author Response

Agradecemos seus comentários